# Microbiology of Diabetic Foot Infections in a Tertiary Care Hospital in São Paulo, Brazil

**DOI:** 10.3390/antibiotics11081125

**Published:** 2022-08-19

**Authors:** Amanda Thurler Palomo, Ana Paula Maia Pires, Marcelo Fernando Matielo, Rafael de Athayde Soares, Christiano Pecego, Roberto Sacilotto, Alexandre Inacio de Paula, Nair Hosino, Cristiano de Melo Gamba, Cibele Lefreve Fonseca, Daniela K. S. Paraskevopoulos, Augusto Yamaguti, João Silva de Mendonça, Silvia Figueiredo Costa, Thaís Guimarães

**Affiliations:** 1Vascular Surgery Department of Hospital do Servidor Público Estadual de São Paulo, São Paulo 04029-000, Brazil; 2Microbiology Department of Hospital do Servidor Público Estadual de São Paulo, São Paulo 04029-000, Brazil; 3Infectious Diseases Department of Hospital do Servidor Público Estadual de São Paulo, São Paulo 04029-000, Brazil; 4Infectious Diseases Department of Hospital das Clínicas, University of São Paulo, São Paulo 05508-220, Brazil

**Keywords:** diabetic foot infections, microbiology, epidemiology

## Abstract

Diabetic foot infections (DFIs) are one of the causes of hospitalization in diabetic patients and, when this occurs, empirical antibiotic therapy is necessary. We have conducted a retrospective study of patients with DFI that required hospitalization to evaluate microbiologic profile and the susceptibility pattern of these infections. We evaluated 320 patients, of which 223 (69.7%) were male with a media age of 71 years with 276 isolates. Gram-positive bacteria were responsible for 188 (68.1%) of the isolates, while Gram-negative bacilli were responsible for 88 (31.9%). *E. faecalis* was the most prevalent pathogen, followed by *S. aureus* and coagulase negative Staphylococci. Among Gram-negative pathogens, *P. aeruginosa* was the most prevalent agent. Regarding the susceptibility profile, we found ampicillin-sensitive enterococci in 89% of the cases, oxacillin-sensitive *S. aureus* in 47%, but in coagulase-negative staphylococci, oxacillin was sensible only in 20%. The susceptibility profile of Gram-negatives was very good with 76% susceptibility of *P. aeruginosa* to ceftazidime and meropenem. The other prevalent Enterobacterales had great susceptibility to ceftazidime, piperacillin-tazobactam and 100% susceptibility to meropenem, with the exception of *K. pneumoniae*, which had 75% susceptibility to meropenem. Knowledge of microbiological profile and susceptibility patterns of patients with DFIs is useful to guide empirical therapy.

## 1. Introduction

The World Health Organization (WHO) published a report in 2016 to mark World Health Day in which it defined diabetes as a global epidemic. The report stated that the number of adults living with diabetes had quadrupled since 1980, reaching 422 million in 2014. This reflected the increases in the risk factors associated with the disease. In 2019, diabetes was the direct cause of 1.5 million deaths and 48% of all deaths due to diabetes occurred before the age of 70 years [1]. The higher number of diabetic patients also brought an increase in the incidence of diabetic foot infections and peripheral arterial disease.

Diabetic foot infections (DFIs) are one of the most common causes of hospitalization in diabetic patients and account for a significant portion of increased hospitalization and healthcare expenditure. These infections are also responsible for the use of broad-spectrum antimicrobials with prolonged duration, which provides the development of bacterial resistance [2].

Foot ulcers start as a result of peripheral neuropathy, which, associated with the decreased neuroendocrine response and sometimes with atherosclerotic peripheral arterial disease, culminates in the appearance of ulcerations and secondary infection [3].

It has already been confirmed that in the overall complexity of DFI, perfusion is just one determinant of the result, while characteristics of the wound and presence and severity of infection are also factors that have a major impact on the risk of limb amputation. As amputation is a very unfavorable event that limits patient functionality and resonates in the social and economic life of society, all efforts must be made to avoid this outcome [4].

There are many classifications available for diabetic foot in trying to estimate the risk of amputation. Previously, existing classification systems for threatened limbs were limited because they do not generally cover all three pillars (wound, ischemia and infection) of the extremity at risk of amputation, nor do they differentiate ulcer from gangrene and so fail to encompass the entire heterogeneous nature of the causes and clinical presentations. Wagner’s classification, which is still widely used for wounds of the diabetic foot, is not much help for differentiating between the causes of ischemic and infectious gangrene [5,6]. Therefore, the Society for Vascular Surgery developed a new classification system that is based on the characteristics of the wound (W), on the degree of ischemia (I) and on the presence and severity of foot infection (fI) called Wound, Ischemia and foot Infection classification, or WIfI classification [7]. Grade IV of WIfI classification estimates 23% amputation risk in 1 year [8].

In cases that require hospitalization, empirical antibiotic therapy is necessary at the moment of diagnosis and is one of the bases for the treatment of diabetic foot [9].

Clinical guidelines from the Infectious Diseases Society of America (IDSA) recommend treating clinically infected diabetic foot ulcers with empirical antibiotics until microbiological culture results are available. Empirical antibiotics should be chosen in accordance with the severity of infection, clinical presentation and the prevalence of microorganisms in the local area and their antibiotic susceptibility. Subsequently, based on daily clinical evaluations and the results of culture and antimicrobial susceptibility test, it may be possible to adjust antimicrobial therapy [10].

On the other hand, broad-spectrum antibiotic regimens can contribute to antibiotic resistance and limit future treatment options. So, to minimize the emergence of bacterial resistance, there is a need to know the local epidemiology, adjust the antimicrobial therapy according to the results of cultures, and avoid prolonged treatments by also thinking about the adverse effects of long-term administration of antibiotics [11].

The purpose of this study was to evaluate the microbiology profile and susceptibility pattern of DFI in patients that require hospitalization.

## 2. Results

We evaluated 320 patients, of which 223 (69.7%) were male with media age of 71 years. The most common comorbidity was systemic arterial hypertension (77.4%), followed by peripheral occlusive arterial disease (57.3%), insulin-dependent diabetes mellitus (38.7%), chronic kidney disease (32.3%), coronary heart disease (15.1%), and smoking (14.8%).

Previous use of antimicrobials within 90 days prior to hospitalization was observed in 87 (27.1%) patients.

Previous amputation and osteomyelitis were both seen in 97 (30.3%) patients. A total of 168 (52.8%) patients had International Working Group on the Diabetic Foot (IWGDF) classification III and 127 (39.8%) had grade IV WIfI classification.

In addition, 276 pathogens were isolated from the 320 patients and Gram-positive bacteria were responsible for 188 (68.1%) of the isolates while Gram-negative bacilli were responsible for 88 (31.9%) of the isolates. Forty-four (13.7%) of the cultures did not have growth of microorganisms. The prevalence of isolated pathogens is described in Table 1.

The susceptibility profile of the Gram-positive and Gram-negative pathogens is described in Table 2 and Table 3, respectively.

## 3. Discussion

We conducted a study to evaluate the prevalence of pathogens and the susceptibility profile of microorganisms isolated from ulcers in diabetic feet infections. This is an essential prevalence study to understand the local epidemiology and thus be able to guide empirical therapy in cases of DFI.

In our study, we found a higher prevalence of males with a mean age of 71 years, denoting the existence of this condition in the elderly population. We found many comorbidities present, with more than half having chronic arterial occlusive disease, which certainly aggravates the circulatory conditions of these patients that can be considered severe arteriopaths.

An epidemiological study conducted in Brazil analyzing 172 diabetic patients demonstrated systemic arterial hypertension present in 84.3% of cases, coronary heart disease in 30.2% of cases, peripheral vascular disease in 8.9%, peripheral neuropathy in 37.9% of cases, foot wound in 10% of cases, amputation in the last 6 months in 6.0% of the cases, hypoglycemia in 21% of cases, ketosis in 2.4% of cases, and infection in 17.1% of cases. This Brazilian cohort was compared with a French cohort that demonstrated statistically significant differences between the two populations, revealing better conditions among the French participants, which highlights the importance of the scientific evidence found in the French study for developing public health actions targeted at older Brazilian people with diabetes mellitus [12].

We also found 52.8% of patients with IWGDF classification considered severe and 39.8% with grade IV WIfI, also severe, where the chance of amputation is about 23% in one year [8]. However, some studies observed that major amputation rates at stage II were the same or even higher than rates at stage III [13,14,15,16]. Recently, a group of researchers from Johns Hopkins Hospital conducted a retrospective analysis of 217 patients with 279 limbs affected by DFI who were seen at their multidisciplinary clinic from 2012 to 2015. They found that the incidence of major amputation at 12 months was similar across WIfI classification stages. They concluded that this classification system was not predictive of risk of major amputation at 1 year for diabetic patients with DFI [17].

Although the risk of amputation may not be estimated by classification systems, the fact is that patients with DFI who require hospitalization are severe, have comorbidities other than diabetes, and have already experienced antimicrobial treatments. In our study, we found 27.1% of patients with previous antimicrobial use. These data were collected in order to try to correlate the use of antimicrobials with bacterial resistance, but this also serves to demonstrate the unsuccessful strategy of outpatient treatment in cases of DFI that evolve to moderate or severe due to the conditions of vascularization of the limbs [18,19].

Another indicator of severity in the population studied is that 30% of patients had previous amputations and osteomyelitis, which is a difficult infection to treat because of antimicrobial bone penetration in patients with arterial disease [20].

The Gram-positive pathogens were in the majority, with *E. faecalis* being the most prevalent pathogen, followed by *S. aureus* and coagulase-negative Staphylococci. Gram-negative pathogens were isolated only in 31.9% of the cases, with *Enterobacterales* being the most prevalent agent accounting for 19.7% of cases, while *P. aeruginosa* was responsible for only 7.6% of cases.

In a recent review that analyzes the global literature relating to incidence, risk factors, resistance patterns and geographic distribution of the microorganisms isolated from diabetic foot infections, *S. aureus* was a significant pathogen, with a growing incidence of *P. aeruginosa* and multi-drug-resistant Gram-negative bacilli [3].

In one study conducted in Indonesia, Gram-negative bacteria were found in 112 (85.5%) subjects, with *Enterobacter* sp. as the predominant bacteria. Gram-positive bacteria were found in 19 (14.5%) subjects, with *Staphylococcus* sp. as the predominant bacteria [21].

Another study conducted in Turkey analyzed the microbiological profile of DFI by conducting a systematic review of articles published over 20 years and compared two periods (1989 to 2007 vs. 2007 to 2011). This identified 31 studies. Overall, these studies reported 2.097 patients, from whom 1.974 microorganisms were isolated. The total percentage of Gram-negative and Gram-positive aerobic bacteria were similar in each of the assessed periods. The rate of isolation of *S. aureus* during the entire period, compared with just the past 5 years was 23.8% and 19.1%, respectively, while the rate of methicillin-resistant *S. aureus* was 7.8% and 5.7%, respectively. The isolation rate of *P. aeruginosa* was 13.7% for the entire period and 14.9% for the past 5 years. While linezolid, vancomycin, and teicoplanin were the most active agents against Gram-positive microorganisms, imipenem and cefoperazone-sulbactam were the most active against Gram-negative microorganisms [22].

On the other hand, the few Brazilian studies published show a high prevalence of the Enterobacterales family (51.5%), with *S. aureus* isolated in 20% and *E. faecalis* in 17.9% of cases [23].

Another Brazilian study found a predominance of Gram-positive bacteria with the most commonly isolated *S. aureus*, followed by *S. saprophyticus*, *S. epidermidis*, *S. agalactiae* and *S. pneumoniae*. Between Gram-negative bacteria, the most commonly isolated were *Proteus* sp. and *Enterobacter* sp., followed by *E. coli*, *Pseudomonas* sp. and *Citrobacter* sp. [24].

While the few Brazilian studies show predominance of *S. aureus*, we found a predominance of Enterococci. These discrepancies demonstrate the importance of knowledge of local epidemiology and resistance patterns that are essential for antibiotic treatment considerations [25].

Regarding the susceptibility profile, we found ampicillin-sensitive enterococci in 89% of the cases and oxacillin-sensitive *S. aureus* in 47% of the cases, but in coagulase-negative staphylococci, the sensitivity to oxacillin was only 20%. It is worth noting that coagulase-negative staphs may be colonizing or contaminating [26]. Although coagulase-negative staphylococci may be colonizing in wounds, in these cases they were considered pathogens because they were collected from deep tissue fragments during surgical procedures [27]. The great susceptibility of enterococci to ampicillin is due to the fact that the vast majority is of the species *faecalis*. As we know, this species has much less resistance when compared to specie *faecium* [28].

A study conducted in Brazil examined 34 individuals with DFI and *S. aureus* was isolated in 17 (50% of cases), of these only 5/17 (29.4%) were resistant to oxacillin [29].

The susceptibility profile of Gram-negatives in our study was very good, with 76% susceptibility of *P. aeruginosa* to ceftazidime and meropenem. The other prevalent Enterobacterales had great susceptibility to ceftazidime, piperacillin-tazobactam and 100% susceptibility to meropenem with the exception of *K. pneumoniae*, which had 75% susceptibility to meropenem.

Ciprofloxacin was the agent with the lowest susceptibility to Gram-negatives, most likely because it is an oral antibiotic, which patients must have used before hospitalization and whose previous use induced resistance. In a Brazilian study conducted in Amazonas, 43.5% of the Gram-negative isolated germs were resistant to ciprofloxacin [23].

In a recent study conducted at China, more than 50% of Gram-negative bacteria were resistant to third-generation cephalosporins, while the resistance rates of piperacillin/tazobactam, amikacin, meropenem, and imipenem were relatively low [30].

The finding of several microorganisms with different susceptibility profiles corroborates the strategy of broad-spectrum therapy with coverage for Gram-positive, Gram-negative and anaerobic pathogens. Coverage for anaerobic pathogens is always indicated because we know their importance in infections where there is a deficit of vascularization and necrosis and the difficulty of isolation of these pathogens in culture due to the conditions of anaerobiosis necessary for their survival [31,32].

Another important issue in the treatment of these infections is the duration of antimicrobial treatment. We know that there is a tendency of prolonged treatments due to the delay of surgical treatment and also a tendency to use antimicrobials until the closure of the lesion. In a randomized, controlled pilot trial, post-debridement antibiotic therapy for soft-tissue DFI for 10 days gave similar (and non-inferior) rates of remission and adverse effects to 20 days [33].

Our study has some limitations. First of all, the retrospective nature of this study precludes the inclusion of information such as glycosylated hemoglobin level, diabetic decompensation, and which antibiotics were taken by patients prior to admission. Moreover, we do not correlate the severity of the diabetic infected foot with the isolated agents. We also did not evaluate the duration of treatment, clinical outcomes or adverse events. This is only a study conducted in a single referral center for the treatment of the elderly and the main objective of this study was to evaluate the prevalence of pathogens and their susceptibility profile.

Despite the limitations mentioned, we are able to implement a new treatment protocol based on the information of the study, where the proposed treatment regimen for cases without previous use of antimicrobials was ampicillin plus ceftazidime plus metronidazole and for cases with previous use of antimicrobials, vancomycin plus piperacillin-tazobactam. The choice for these regimens was due to the high prevalence of ampicillin-sensitive Enterococci and ceftazidime-sensitive *Pseudomonas aeruginosa*. Anaerobic cover is always added to cases of DFI. For cases using previous antibiotic therapy, vancomycin was added to cover Staphylococci as coagulase-negative and *S. aureus* were 80% and 54% resistant to oxacillin, respectively. Piperacillin-tazobactam was added for Gram-negative and anaerobic pathogen coverage.

This strategy of guiding empirical therapy based on local epidemiology is very important to educate prescribers in the management of infections, to contribute to improve knowledge about bacterial resistance, and to implement antimicrobial stewardship practices [34].

In conclusion, knowledge of microbiological profile and susceptibility patterns of patients with DFI is useful to guide empirical therapy in these serious infections that are difficult to treat and that require hospitalization to avoid amputations and to improve the living conditions of these patients. Nevertheless, each hospital should obtain its own microbiologic profile of patients with diabetic foot infections and treat them with the most appropriate empirical antibiotic.

## 4. Methods

This is a retrospective study conducted through the analysis of medical records of patients with diabetic foot infections that required hospitalization from January 2017 to December 2019 at Hospital do Servidor Público Estadual (HSPE) de São Paulo, located in São Paulo, Brazil.

The HSPE is a tertiary teaching hospital, with 823 beds and 77 beds for intensive care units. It also has clinical and surgical hospitalization wards of various specialties. The Department of Vascular Surgery has an inpatient unit with 24 beds for the treatment of vascular diseases and DFI is responsible for about 40% of hospitalized cases.

The inclusion criteria were all patients hospitalized with DFI who required debridement. All debridements were performed in an operating room environment with aseptic techniques, and fragments of tendon or deep tissue were removed and sent for culture in a sterile dry tube. Upon receiving the material, the microbiology laboratory sowed the fragments in blood agar and chocolate agar or medium containing thioglycolate. After growth, bacterial colonies were identified and submitted to a susceptibility test in an automated Phoenix^TM^^®^ (Becton-Dickinson, São Paulo, Brazil, 2015).

The variables studied were: demographic data; the presence of comorbidities; previous use of antimicrobials until 90 days before; previous amputation; the presence of osteomyelitis diagnosed by imaging; and identified etiologic agent and susceptibility test.

All patients were classified according to classification of diabetic foot ulcers by International Working Group on the Diabetic Foot (IWGDF) and by a system using risk stratification based on wound, ischemia and foot infection (WIfI) [6,7].

All information regarding patients was stored in a database using the Microsoft Excel version 5.0 for Windows (São Paulo, Brazil, 1999) program and frequency analysis, and percentage was also estimated.

## Figures and Tables

**Table 1 antibiotics-11-01125-t001:** Isolated pathogens in patients admitted with DFI (*n* = 276).

Gram-Positive	*n*	%	Gram-Negative	*n*	%
*E. faecalis*	68	24.6	*Pseudomonas aeruginosa*	21	7.6
*S. aureus*	47	17.0	*Proteus* sp.	13	4.7
*Other* Coagulase-negative *Staphylococci*	40	14.4	Other *Enterobacterales*	13	4.7
*S. epidermidis*	21	7.6	*Klebsiella pneumoniae*	12	4.3
*E. faecium*	5	1.8	*E. coli*	9	3.2
*Enterococcus* sp.	4	1.4	*Enterobacter* sp.	8	2.8
*S. agalactiae*	3	1.0	*Acinetobacter* sp.	6	2.1
			*Stenotrophomonas maltophilia*	5	1.8
			*Burkolderia cepacia*	1	0.3

**Table 2 antibiotics-11-01125-t002:** Susceptibility pattern of the Gram-positive pathogens.

Pathogen (N)		
	Susceptibility to oxacillin	
	N	%
*S. aureus* (44)	21	47.7
*S. epidermidis* (20)	3	15.0
Other Coagulase-negative *Staphylococci* (36)	2	5.5
	Susceptibility to ampicillin	
*Enterococcus faecalis* (64)	57	89
*Enterococcus faecium* (5)	0	0
*Enterococcus* sp. (3)	3	100

**Table 3 antibiotics-11-01125-t003:** Susceptibility pattern of the Gram-negative pathogens.

Pathogen (N)	Susceptibility											
	Ciprofloxacin		Ceftriaxone		Ceftazidime		Cefepime		Piperacillin-Tazobactam		Meropenem	
	N	%	N	%	N	%	N	%	N	%	N	%
*P. aeruginosa* (21)	10	47.6	-	-	16	76.1	15	71.4	12	57.1	16	76.1
*Acinetobacter* sp. (6)	3	50	-	-	3	50	3	50	2	33.3	3	50
*Proteus* sp. (13)	8	61.5	7	53.8	7	53.8	11	84.6	11	84.6	13	100
Others *Enterobacterales* (13)	6	46.1	7	53.8	7	53.8	8	61.5	11	84.6	13	100
*Klebsiella pneumoniae* (12)	4	33.3	5	41.6	5	41.6	6	50	6	50	9	75
*E. coli* (9)	5	55.5	7	77.7	7	77.7	8	88.8	9	100	9	100
*Enterobacter* sp. (8)	5	62.5	4	50	4	50	5	62.5	8	100	8	100

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
