# Peer review of "Microbiology of Diabetic Foot Infections in a Tertiary Care Hospital in São Paulo, Brazil"

_antibiotics, 2022, doi:10.3390/antibiotics11081125_

Round 1

Reviewer 1 Report

Many thanks for the opportunity to reviewer this interesting paper. 

Diabetic foot infection is an underestimated problem, and in my opinion the paper is suitable of publication after minor revision

Introduction: updata data on global report of diabetes and introduce better why diabetic ulcer infection is an important problem (elderly patients, comorbidities, fragile patients, length of hospitalitation, cost...). Furthermore, closely with diabetic foot infection is topic of antibiotic resistance that continually spread wordwilde also during COVID 19 pandemic (see and cite Impact of SARS-CoV-2 Epidemic on Antimicrobial Resistance: A Literature Review. Viruses. 2021 Oct 20;13(11):2110. doi: 10.3390/v13112110.)

Methods: clear, please describe that if germs are derived from biopsies or skin swabs

Resutls: any data on outcome? hospitalization time? if no put it in limitations section

Discussion: please add one issue in my opinion very relevan: the education on antimicrobial resistance (AMR) /appropiate antibiotic therapy. For example the introduction of a course on AMR  during medicine and nurse degree could be a strategy for improve knowledge and awareness on this topic and improve the manage of spread of antibiotic resistance (see and cite Italian young doctors' knowledge, attitudes and practices on antibiotic use and resistance: A national cross-sectional survey. J Glob Antimicrob Resist. 2020 Dec;23:167-173. doi: 10.1016/j.jgar.2020.08.022)

Moreover , add limitations in your paper (no data on outcome, no data on therapy...) but it is an important paper on the epidemiology of infection

in addition give some public health conclusion that came from your paper and your proposal to contain and better manage the problem

Reviewer 2 Report

The authors aimed to share their study regarding microbiology in diabetic food infections in their hospital. I appreciate their efforts in helping their patients, collecting specimens, and isolating various bacterial strains to test their susceptibility. The reason for this study is mentioned - each hospital should have its own microbiological profile for DGI patients to ensure the most appropriate treatment, combining the medical staff's observations with the current international therapeutic protocols.

The other comments and suggestions are in the attached manuscript.
